# Source Apportionment of Fine Organic Particulate Matter (PM_2.5_) in Central Addis Ababa, Ethiopia

**DOI:** 10.3390/ijerph182111608

**Published:** 2021-11-04

**Authors:** Worku Tefera, Abera Kumie, Kiros Berhane, Frank Gilliland, Alexandra Lai, Piyaporn Sricharoenvech, Jonathan Patz, Jonathan Samet, James J. Schauer

**Affiliations:** 1School of Public Health, College of Health Sciences, Addis Ababa University, Addis Ababa 9086, Ethiopia; aberakumie2@yahoo.com or; 2Department of Biostatistics, Columbia University, New York, NY 10032, USA; ktb2132@cumc.columbia.edu; 3Keck School of Medicine, University of Southern California, Los Angeles, CA 90033, USA; gillilan@usc.edu; 4Environmental Chemistry and Technology Program, University of Wisconsin-Madison, Madison, WI 53706, USA; alexandra.lai@weizmann.ac.il (A.L.); sricharoenve@wisc.edu (P.S.); jjschauer@wisc.edu (J.J.S.); 5Global Health Institute, University of Wisconsin, Madison, WI 53706, USA; patz@wisc.edu; 6Office of the Dean, Colorado School of Public Health, Aurora, CO 80045, USA; jon.samet@CUAnschutz.edu; 7Wisconsin State Laboratory of Hygiene, University of Wisconsin-Madison, Madison, WI 53706, USA

**Keywords:** ambient air pollution, motor vehicles, biomass burning, soil dust, seasonality, source apportionment, chemical mass balance (CMB)

## Abstract

The development of infrastructure, a rapidly increasing population, and urbanization has resulted in increasing air pollution levels in the African city of Addis Ababa. Prior investigations into air pollution have not yet sufficiently addressed the sources of atmospheric particulate matter. This study aims to identify the major sources of fine particulate matter (PM_2.5_) and its seasonal contribution in Addis Ababa, Ethiopia. Twenty-four-hour average PM_2.5_ mass samples were collected every 6th day, from November 2015 through November 2016. Chemical species were measured in samples and source apportionment was conducted using a chemical mass balance (CMB) receptor model that uses particle-phase organic tracer concentrations to estimate source contributions to PM_2.5_ organic carbon (OC) and the overall PM_2.5_ mass. Vehicular sources (28%), biomass burning (18.3%), plus soil dust (17.4%) comprise about two-thirds of the PM_2.5_ mass, followed by sulfate (6.5%). The sources of air pollution vary seasonally, particularly during the main wet season (June–September) and short rain season (February–April): From motor vehicles, (31.0 ± 2.6%) vs. (24.7 ± 1.2%); biomass burning, (21.5 ± 5%) vs. (14 ± 2%); and soil dust, (11 ± 6.4%) vs. (22.7 ± 8.4%), respectively, are amongst the three principal sources of ambient PM_2.5_ mass in the city. We suggest policy measures focusing on transportation, cleaner fuel or energy, waste management, and increasing awareness on the impact of air pollution on the public’s health.

## 1. Introduction

Major cities in Africa, such as Addis Ababa, experience a substantial disease burden from high levels of ambient air pollution that reflect rising economic activities, a growing population, and rapid urbanization [1,2]. This substantial disease burden largely comes from airborne particulate matter (PM), linked causally to increased morbidity and premature mortality. However, the worsening air pollution problem is not well tracked across sub-Saharan Africa, as real-time continuous air quality monitoring is not in place in most African cities [3,4]. There has not been a sufficient characterization of sources and their contribution to ambient PM, as only a few source apportionment studies have been undertaken to-date in Africa [3,5,6,7,8,9,10,11].

Published and unpublished data from Ethiopia has shown that ambient fine PM (PM_2.5_—PM ≤ 2.5 microns in aerodynamic diameter) are high and above the Air Quality Guidelines (AQG) of the World Health Organization (WHO) [3,4,12]. Prior studies, for instance, in Addis Ababa [5,13] reported that PM_10_ concentrations were increasing and above the annual value of WHO air quality guidelines [14] and Ethiopian standards [15]. However, only a few studies characterizing the sources of ambient air pollution have been carried out in large cities of sub-Saharan Africa where ambient air pollution appears to be worsening. The studies carried out to date show that major sources of air pollution include biomass burning, traffic sources, industries, and dust [6,8,9,10] (see Appendix A).

Here, we report the findings of a source apportionment study carried out in Addis Ababa, the capital of Ethiopia, with a population of over 4.7 million. Monitoring data from the nearby US Embassy show that PM_2.5_ levels are well above WHO AQG values for 24-h and annual means—WHO AQG (2021) [16]; 15 and 5 μg.m^−3^; and EPA Ethiopia AQ standard (2003) [15]; 65 and 15 μg m^−3^), respectively. The city has increasing vehicle traffic with poorly-controlled emissions and substantial numbers of diesel vehicles without controls [17,18,19]. Biomass burning is widespread and the most common energy source for cooking [20]. Only two source apportionment studies have been carried out on PM_10_ mass in the city so far, but none were published during the last decade including on PM_2.5_ mass [5,13]. The present study, based on data collected at a single site in 2015–2016 [12], describes air pollution sources based on chemical mass balance analysis.

## 2. Materials and Methods

### 2.1. Sampling Site Description

We conducted this study longitudinally over a one-year period in Addis Ababa. The selected sampling site (Geo-reference: 9.019046N, 038.747361E; Altitude: 2355 masl) was at a meteorological station, near the city’s center with instrument placement at 2 m above the ground. The sampling site was devoid of large trees and surrounded by offices, including a three-story building at 28 m from the sampler, towards the west. The closest busy roadway (primary arterial street (PAS)) was at a distance of 536 m while a secondary arterial street (SAS) and a connecting street were at 64 m and 25 m, respectively. The United States Environmental Protection Agency (US EPA) near road sampling recommendation for ambient air quality collocation monitoring is 1–10 m [21]. The prevailing winds blow from easterly and southeasterly directions, implying that the pollution from motor vehicles, household biomass-burning, industries, and other sources are likely pushed towards the western direction, across the site. The ambient temperature of Addis Ababa city ranges largely from 8.9 °C (48 °F) to 23.9 °C (75 °F) and is rarely below 5.5 °C (42 °F) or above 26.7 °C (80 °F) extremities [22].

### 2.2. Sampling Method

We deployed three Harvard Impactor (HI) (Boston, MA, USA, 5 L/min) 5-stage cascade samplers [23] at the site to measure PM_2.5_ mass on 24-h integrated filters. We collected samples from November 2015 to November 2016 once, every sixth day from 00:00 local time (mid-night). One Harvard Impactor sampler, filled with a Teflon fiber filter (37 mm PTFE, Whatman, NJ, USA), was used to measure fine PM and trace metals, and another with a quartz filter (37 mm, quartz fiber 2500QAT-UP, Pall Life Sciences, Westborough, MA, USA) for chemical speciation. The third filter was used as a co-located back up. Field blanks comprised of 9 filters (8 Teflon and 1 quartz). After collection, we stored the filters in petri dishes and placed them in a refrigerator, at −20 °C pending analyses.

All quartz filters were analyzed for organic carbon (OC) and elemental carbon (EC) using a Sunset Laboratory analyzer as described by Watson et al. (1997) [21]. Depending on the number of samples collected each month of every 6th day schedule, five or six samples collected within each month were aggregated to get mass loading that is commensurate for Gas Chromatography-Mass Spectrometry (GC-MS) analyses including organic tracers. The sampling strategy is commonly used for aerosol chemistry studies to strike a balance of maintaining representativeness with pragmatic use of sampling and analysis resources. We analyzed the samples we had as composites. Five days, 24 December (2015), 5 May, 9 and 15 June, and 3 July (2016), were missed during sampling.

### 2.3. Chemical Analysis

The sampling approach and the settings are described in more detail by Tefera et al. (2020) [12]. X-ray fluorescence was used for elemental analyses. The Thermal Optical Analyzer (Sunset Laboratories, Forest Grove, OR, USA) was used to measure EC/OC with the National Institute of Occupational Safety and Health (NIOSH) thermal optical transmission method [24].

Schauer (2003) [25] described the details on the methods of EC and OC analyses. As described in detail by Tefera et al. (2020) [12], water-soluble organic carbon (WSOC) and water-insoluble organic carbon were determined following total organic carbon (TOC) analyses [26]. Ion chromatography (IC) (Dionex ICS 2100 and Dionex ICS 100) measured these seven ions: Sulfate (SO_4_^2−^), nitrate (NO_3_^−^), chloride (Cl^−^), sodium (Na^+^), ammonium (NH_4_^+^), potassium (K^+^), and calcium (Ca^2+^) [27].

Composites were prepared on monthly basis to analyze organic compounds using Gas Chromatography-Mass Spectrometry (GC-6980, quadrupole MS-5973, Agilent Technology, Santa Clara, CA, USA). Each monthly composite contained equal portions of each of the 5 or 6 quartz filters collected over that month and contained at least 500 µg of OC to ensure the detection of organic compounds by GC-MS. Before extraction, isotopically labeled standard solutions were used to spike the composites. A 50/50 dichloromethane and acetone mixture were used to extract the composite samples. Samples were sonicated in 20-min increments, alternating between the solvents for a total of four successive extractions. The resulting extracts were then reduced in volume using a rotary evaporator followed by evaporation under ultra-pure nitrogen. We analyzed the extracts twice, using the GC-MS. One aliquot of each was methylated with diazomethane to convert organic acids into methyl esters before analysis and a second aliquot was derivatized with a silylating reagent for analysis of polar oxygenated organic compounds including levoglucosan and cholesterol [28].

Quality Assurance and Quality Control (QA/QC) carried out as described in prior studies [29,30]. The concentration of each chemical species was blank-corrected, and the standard deviation of the field blanks and the detection limit of the instruments were used to estimate the uncertainties. For the GC-MS data, the spike recoveries were used when propagating the uncertainty of the concentrations.

### 2.4. Source Apportionment for PM_2.5_ OC by CMB

Chemical mass balance (CMB) model (v.8.2) was used to quantify the primary source contributions for PM_2.5_ organic carbon. This version is the latest receptor model (https://www.epa.gov/scram/chemical-mass-balance-cmb-model, accessed on 29 September 2021). The CMB model calculates source contributions from inputs of source profiles and ambient concentrations of organic tracers by solving a system of linear equations using the effective variance weighted least-square method [31]. Data from the monthly composite samples were used to run the CMB model, as GC-MS analysis of individual daily samples was not feasible due to limited PM_2.5_ mass. Organic molecular markers selected as model inputs did not react or volatilize and hence were stable during transport from the location of the sample until the laboratory. Molecular markers were also selected to correspond to principal sources of pollution, represented by the source profiles included in the CMB model [32].

The CMB model used 13 molecular markers including EC, picene, hopanes (17α(H)-22,29,30-trisnorhopane, 17α(H)-21β(H)-30-norhopane, 17α(H)-21β(H)-hopane), ABB-20R-C27-cholestane, ABB-20R-C29-sitostane, ABB-20S-C29-sitostane, n-alkanes with carbon numbers from 27 to 33, and levoglucosan. Finally, findings from studies in other African cities led to the choice of four source profiles (see Appendix A). Hence, we selected biomass fuel [33], motor vehicles [34], residential coal burning [35], and fuel oil [35,36,37] as source candidates. CMB results with a coefficient of determination—R^2^ when less than 0.8; the estimates of source contributions do not explain the observations under study whereas when R^2^ is closer to 1.0 better explains the observations’ source contributions [38]. Sensitivity analysis was applied to control collinearity problems with the sources to select the pertinent major sources of OC [32].

## 3. Results and Discussion

### 3.1. Organic Tracers and Alkanes

#### 3.1.1. Fine PM Organic Tracers

Gas Chromatography Mass Spectrometry (GC-MS) analysis of the samples yielded 126 organic species. Organic tracers contribute to PM_2.5_ mass at very low concentration levels, yet their role in identifying and quantifying sources of atmospheric particulate matter is critical. The monthly distributions of the concentrations of levoglucosan, hopanes, and picene showed similar seasonal patterns (Figure 1a–c).

Biomass burning represented via a robust organic molecular marker—levoglucosan [39]—showed the highest and lowest concentrations during June and July (average, 2234 ± 204 ng m^−3^) and during February (619 ng m^−3^), respectively, in which the average lowest concentration was 3.6 times less than that of the highest months.

The wet seasons showed high concentrations of levoglucosan. Massive solid biomass used for cooking and space heating, and temperature inversions are likely responsible.

Findings were similar to a study in four neighborhoods of Accra, Ghana, which found a significant contribution of biomass burning to PM_2.5_ mass [3]. Other studies had similar findings in urban areas of SSA with biomass burning [40].

Incomplete fossil fuel combustion and burning of vegetation at high temperatures forms polycyclic aromatic hydrocarbons (PAH), known carcinogens that include benzo[a]pyrene, benzo[b]fluoranthene, benzo[e]purene, and benzo[k]fluoranthene. While some other studies included PAHs as organic tracers for gasoline and wood combustion [41], the present study did not include PAHs in the CMB model.

Hopanes are important organic markers for traffic-related emissions [42], and hopanes come mainly from the motor exhaust of diesel and gasoline vehicles and the combustion of fuel oil. Figure 1b shows the monthly concentrations of the measured hopanes, namely 17α(H)-22,29,30-trisnorhopane, 17α(H)-21β(H)-30-norhopane, and 17α(H)-21β(H)-hopane. A seasonal pattern was observed in concentrations of hopanes with higher concentrations during the wet months, especially in June and July. Concentrations of these three hopanes in total range from the lowest in May (1.132 ng m^−3^) to the highest in July (3.539 ng m^−3^) and an overall average of 1.955 ± 0.70 ng m^−3^. A study in Santiago had a similar finding with higher concentrations of hopanes during cold seasons of the year [43]. Emission sources, although not specified in this study, can be diesel and gasoline and oil combustion, and concentrations increased by temperature inversion and low dispersion during this season.

Picene is a polycyclic hydrocarbon, used in the samples as an organic marker for coal combustion [44]. The concentration of picene ranged from a high of 1.7 ng m^−3^ (July) to a low of 0.3 ng m^−3^ (October), averaging 0.9 ± 0.3 ng m^−3^. The seasonal pattern of picene concentration follows the ambient temperature patterns, reaching high levels during the wet seasons of the year as opposed to dry seasons in October and May, characterized by low precipitation and a relatively higher temperature. Notably, picene concentrations rose when residential coal is most used and atmospheric dispersion conditions reduced by thermal inversions. Unlike levoglucosan, organic tracers such as hopanes and picene have very low, but detectable concentrations. However, all three groups of organic tracers have a similar seasonal pattern (Figure 1a–c).

#### 3.1.2. Concentration of C27-C33n-alkanes

Concentrations of C28-31 n-alkanes (n-Octacosane, Nonacosane, Triacontane, and Hentriacontane) in Appendix A–c show a seasonal pattern similar to those for the organic tracers mentioned above. The sum of all four n-alkanes ranged from a low of 39.10 ng m^−3^ in May to a high of 98.21 ng m^−3^ in July, averaging 60.01 ± 16.91 ng m^−3^. The carbon preference index (CPI), which is expressed as the ratio of odd to even n-alkanes, was calculated to determine the origins of n-alkanes (C28-31), which include biogenic detritus such as plant wax and microbes, and anthropogenic emissions such as oils, soot, and synthetics [39,45].

CPI values approaching 1.0 indicate emissions from fossil fuel while values above 2.0 to 10.0 indicate that biogenic detritus is the origin of n-alkanes [36,45,46,47]. CPI values in this study ranged from a low of 0.6 (October) to a high of 1.7 (January), with an average CPI value of 1.3 (Appendix A, which may show anthropogenic sources resulted in the n-alkanes. The lower the value of the CPI, the lighter the fuel oil and the higher the maturity of the oil (see Appendix A). Heavy and immature crude contains petrochemicals that might have detrimental effects on petroleum operations, including transportation [48].

### 3.2. CMB Result for Sources of fine PM and Its OC Components

#### 3.2.1. Source Apportionment for OC

The chemical mass balance (CMB) model, a receptor model, was used to estimate source contributions to OC from biomass burning, residential coal combustion, fuel oil, and mobile sources, selected based on organic molecular tracer concentrations and sensitivity analyses. “Other OC” defined as the difference between TOC and the sum of the contributions of all primary sources of OC incorporated in the receptor model, secondary organic aerosols (SOA), and other unidentified primary sources of OC. Appendix A show the contributions from the OC sources along with their uncertainties by mass concentrations and proportions, respectively. Figure 2 shows the monthly contributions of sources to OC by mass concentrations and percentage, including solely those sources yielding statistically significant contributions.

The CMB model included two mobile sources (vehicles and fuel oil), but the model does not distinguish motor vehicle sources by fuel type—diesel and gasoline. Using motor vehicle category minimizes this uncertainty compared to the split mobile sources, which likely have a larger impact on source apportionment estimates of diesel and gasoline [34]. The seasonal pattern of mobile source emissions, with statistically significant contributions, agrees well with the total PM_2.5_ mass patterns. The model fits well for the mobile sources category, with a high coefficient of determination (R^2^ > 0.90).

Over the period of May–November 2016, the largest contributor to OC was biomass burning, while vehicles were the second-largest contributor to OC, while being the dominant source for the total PM_2.5_ mass. The contributions of both sources (biomass burning and vehicles) were much lower during the preceding period (November 2015–April 2016). Biomass burning in wet and colder seasons (May–November 2016) contributed 13.0 ± 2.6 μg m^−3^, and was particularly important during June and July, when it contributed 15.2 ± 3.1 μg m^−3^, which accounts for 61 ± 15% of the mass in OC.

The vehicles were the second-most important primary source that contributes from a low of 3.7 μg m^−3^ (April) to a high of 10.5 μg m^−3^ (June), bringing a significantly increased OC contribution, generally during the rainy and cold months (May–November 2016). These values average 47 ± 9% during the period of May–November compared with the annual vs. dry and short rain season estimated mean contributions, averaging 37 ± 8% vs. 25 ± 7%, respectively. The higher OC concentrations during the rainy season were due to the significantly higher emissions rates in this season [12,49], which resulted in higher concentrations even with the higher deposition [50,51] due to rain (See Figure 2a). A recently reported study elsewhere by Zhang et al. (2020) [52] similarly shows that in colder seasons, residential heating contributes large amounts of OC to TOC. Fuel oil contributed the lowest OC during the year, accounting for an overall average of 0.012 ± 0.02 μg m^−3^, which is just 0.07 ± 0.02% of OC.

In contrast, most estimates for “Other OC” contributions were negative, or the concentrations were less than twice the uncertainty (i.e., not statistically significant), except for the three months of November 2015, and February and March 2016, during which the statistically significant “Other OC” changes from 7.1 μg m^−3^ in November 2015 to 9.4 μg m^−3^ and 5.9 μg m^−3^ in February and March 2016, respectively. Thus, “Other OC” might make predominant contributions to OC during the warmer months (e.g., February accounts for 49% of OC). The relatively high proportion of contributions of “Other OC” to OC is likely comprised of SOA, the formation of which by photochemical oxidation may dominate aqueous-phase processes, during warmer months.

The relative impact of sources of air pollution arising from fine particulate organic carbon can be predicted by examining the source tracers’ seasonal pattern (see Figure 1a–c and Figure 2a,b)). Figure 3a,b also show the source apportionment analysis results. The results from the raw data and source apportionment show the consistency of the findings, which is an important measure of consistency check and suggests confidence in the apportionment of the sources. The levels in levoglucosan vary with seasonality, which is characterized by wet or colder months that increases the source-tracer of biomass burning for cooking and space heating. In the cold months, poor dispersion conditions following temperature inversionse [5] and decreased photochemical reactions affect the concentration of emissions for these sources, which has not been considered in the present study. Therefore, seasonality is more marked with levoglucosan than the tracers for motor vehicles. Yet, dispersion plays a critical role for determining the pollution concentration from these sources.

#### 3.2.2. Source Apportionment for Fine PM Mass

The source contributions to OC, as determined by the chemical mass balance (CMB), are converted to that of fine PM mass via specific factors (OC/PM_2.5_) for each of the identified sources [33,53]. The selection of source profiles was based on an analysis of the organic molecular markers measured in the study to determine the sources that were most important in Addis Ababa. This approach has been widely used on source apportionment studies in North America, Asia, Europe, the Middle East, and Africa. The fact that the source profiles are based on normalized composition, and not absolute emissions rates, allows the source profiles to be more universal in nature.

As described in Tefera et al. (2020) [12], we used a factor of 1.4 for OM:OC ratio to convert OC to OM. However, as Brown et al. (2013) [54] investigated, variability pertaining to the OM/OC ratio on a daily or a seasonal basis. Hence, the use of a constant factor challenges the validity of OM estimates for different OM sources under natural environment conditions. It, therefore, warrants on cautious interpretation of health impact estimates from the particulate OM sources. In addition, sources including sulfate, nitrate, and ammonium concentrations input into CMB. Prior studies in other locales estimated SOA by converting the “CMB Other” OC to Organic Mass by using a factor of 2.0 to account for the oxygen and hydrogen that consisted in OM but not OC [43,55]. The overall average SOA estimate in this study is 7.04 μg m^−3^. We subtract the product of the minimum ratio of ambient OC to EC with ambient EC from the total OC to yield SOA [56].

The undetermined mass, referred to as “Other PM_2.5_”, calculated as the difference between the sums of all sources of PM_2.5_ mass determined using the CMB and the gravimetric PM_2.5_ mass. Data from Appendix A, and Figure 3 shows the contributions of these sources to PM_2.5_ mass.

Motor vehicles make a major source of primary PM_2.5_ mass across the entire year, with average contributions of 14.0 ± 2.3 μg m^−3^ (28 ± 9.1%). The highest contribution from vehicles was during September 2016–November 2016, averaging 16.7 ± 2.6 μg m^−3^ (36.3 ± 9.6%). Vehicles emit important levels of nitrogen oxides but do not emit significant amounts of direct emissions of nitrate aerosol. The nitrogen oxide emissions will oxidize in the atmosphere to nitrate aerosol but in the case of Addis Ababa, the chemical conversion occurs downwind of the city, which explains the low concentrations of nitrate aerosol at the sampling site. The contribution of biomass burning ranges from a low of 5.0 μg m^−3^ (February) to a high of 19.3 μg m^−3^ (July). Fuel oil contributed the least mass, amongst identified sources, to PM_2.5_ mass, with an average of 0.31 ± 0.06 μg m^−3^, which is less than 1%.

The most important secondary sources contributed to sulfate, averaging 3.06 ± 0.49 μg m^−3^ (6.5 ± 2.2%). Whereas sulfate negatively related with biomass burning, a positive correlation observed with vehicles, suggesting that the high sulfur fuel sold in the city might have led to this level [19]. The linearly increasing trend of percentage contribution of vehicles to PM_2.5_ mass, suggesting that given the trend of change, its contribution grew by a monthly increment factor of 1.41, totaling an increment of 16.8% per annum (coefficient of determination, R^2^ = 0.8095, and model intercept = 0.1816). This rate of change is consistent with the annual rate of increase in the motor vehicular fleet in the city [18].

Secondary inorganic aerosols, comprising of sulfate, ammonium, and nitrate, in total accounted for 12.6 ± 3.1% of PM_2.5_ mass from September to November (2016). “Other PM_2.5_” mass, which contributed to 22.7% of the total PM_2.5_ mass, varied from the highest in August (43%) to the lowest in May (1%). However, there is no identified seasonal pattern observed. Except for June, none of the months showed a significant contribution to the fine PM total mass.

## 4. Conclusions

The ambient aerosols samples were collected from November 2015 to November 2016 at a city center location in the city of Addis Ababa. A receptor model—chemical mass balance—was used in which molecular markers: EC|OC, hopanes, picene, and levoglucosan, were included in the model. The CMB model determined the following sources and the respective contributions to OC and fine PM: Biomass burning, motor vehicles, residential coal, and fuel oil.

The ambient aerosols were highly affected by primary sources from motor vehicles and biomass combustion, besides secondary aerosols, which are formed in the atmosphere. Motor vehicles contributed an average of 37% (ranging from 23 to 56%) of ambient OC. The contribution of motor vehicles to PM_2.5_ mass was fairly consistent and accounted for 18–37% of PM_2.5_ mass. The biomass combustion contributions to ambient OC and PM_2.5_ mass identified as an important source, with an average of 46 and 18%, which ranges from 22 to 74% and 12 to 28%, respectively.

The patterns of the concentration of the molecular markers were the same as that of PM_2.5_ mass. The high concentrations were recorded during June to September, particularly in June and July. Biomass burning contributed the most to OC, and it was especially high in June, July, and November (2016). June–November 2016 had higher levels (over 50%). The contribution of motor vehicles exceeded that of biomass-burning throughout the year especially during the months of June, July, and November where the latter exceled by 15–24% more than motor vehicle sources, yet during February and October, motor vehicles’ contribution margins over biomass sources were larger. “Other OC” contributions to OC recorded high levels during the dry and short rain seasons (November, February, and March); the highest was 49% in February. The seasonal patterns of “Other OC” suggest that most of these could arise from SOA because of photochemical reactions.

Motor vehicles are a primary source of PM_2.5_ mass. The motor vehicles source contribution was highest in June 2016, reaching 24.07 ± 2.7 μg m^−3^, amounting to 29 ± 4% of PM_2.5_ mass; yet the highest contributions by proportion were in September and November, which represented 37 ± 10% (22.48 ± 3.36 μg m^−3^) and 37 ± 15.5% (14.61 ± 2.59 μg m^−3^), respectively. The second-highest primary source contributor to PM_2.5_ mass was biomass-burning, which was especially high during the main rainy season (June–September), averaging 15.5 ± 3.12 μg m^−3^, which contributes to the highest level in July, 28 ± 9% (19.32 ± 3.88 μg m^−3^).

Sulfate, averaging 6.5%, contributes the highest amongst the secondary inorganic ions while ammonium (2.0%) and nitrate (0.6%) contributes of PM_2.5_ mass from November 2015–November 2016. Sulfate ions apparently come from the high sulfur content diesel in fuel from sources of vehicles and from diesel generators as the industrial contribution appears may be at its lower stage yet. The low nitrate level in the face of motor vehicles being a major contributor to PM_2.5_ mass might seem paradoxical to the relatively lower number of fleets in Addis Ababa compared to major cities elsewhere in other countries, and thus, this needs further investigation.

Reconstructed soil dust materials, derived from oxides of metals, contributed 17.4 ± 6.6%, and had a similar proportion to a biomass burning. The level of soil dust was more prevalent during the dry and short rain seasons of May, February and March 2016, averaging 30.7 ± 11%, the highest level was a day in May (37 ± 20%). This study estimates SOA contributions are 7.04 μg m^−3^, yet the method of estimation constrained by assumptions and may introduce some error [56]. Undetermined mass accounted for 22.7 ± 31.2% of PM_2.5_ mass. Data during June 2016 showed a significant level of “Other PM” concentrations, contributing 21 ± 10% of PM_2.5_ mass.

In summary, we can conclude from the results that the three most abundant sources—motor vehicle, biomass burning, and soil dust—accounted for nearly 2/3rd of the PM_2.5_ mass concentration. There is lower seasonal variability in their contribution. The implication of these findings is that interventions focusing on these sources gaged by seasonality could affect the reduction of PM_2.5_ mass and hence, improve the health of dwellers and the economy. This result calls for the government of Ethiopia to constrain these sources to enable the reduction of air pollution and improve air quality, via policy and regulatory interventions focusing on importing used cars (putting tighter age-cap), reducing sulfur content of imported fuel, and putting more stringent motor vehicle emission standards and enforcement of regulation, and shifting household cooking energy mix from solid fuel to renewable and clean alternatives. A policy that holds polluting cars accountable through taxation and issuing stricter enforceable emission standards may help reduce pollution in the city.

## Figures and Tables

**Figure 1 ijerph-18-11608-f001:**
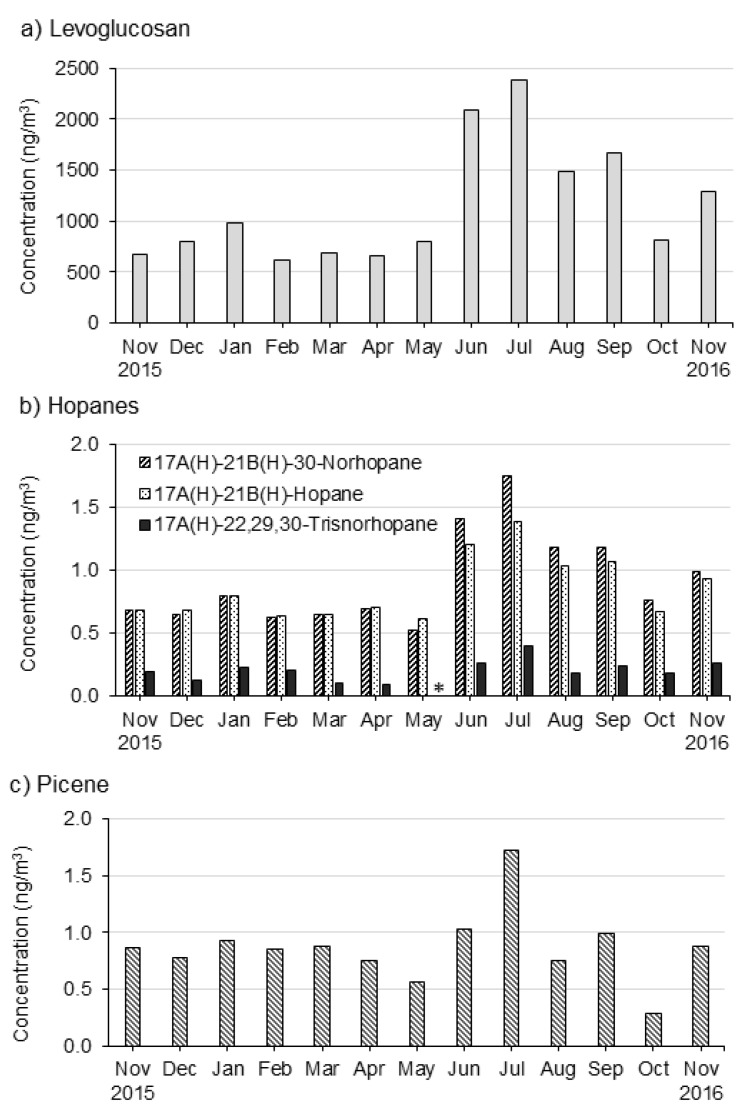
Monthly trends in concentrations of selected organic tracers used as fitting species in the chemical mass balance model: (**a**) Levoglucosan, (**b**) Hopanes, and (**c**) Picene. * indicates below detection.

**Figure 2 ijerph-18-11608-f002:**
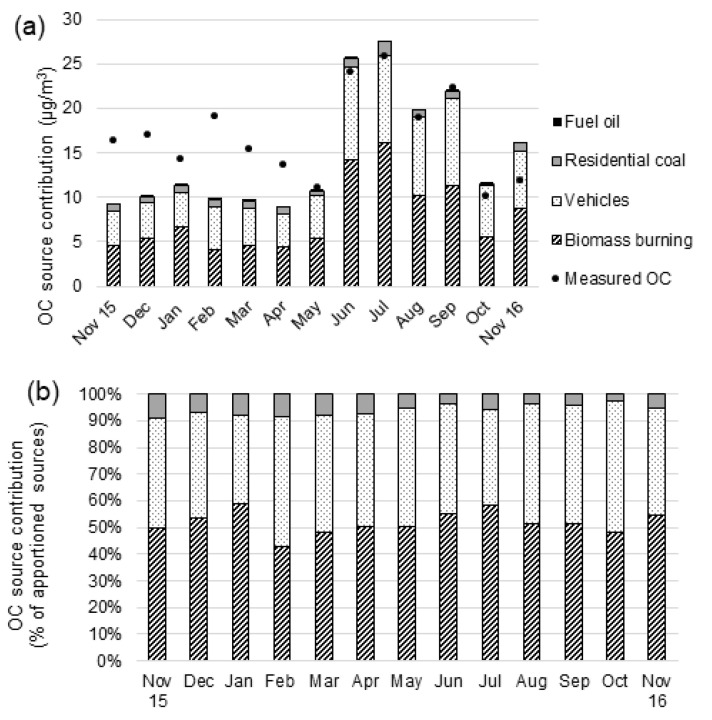
Chemical mass balance (CMB) organic carbon (OC) source contribution estimates in units of (**a**) µg.m^−3^ and (**b**) percentage of apportioned OC in Addis Ababa, November 2015 to November 2016.

**Figure 3 ijerph-18-11608-f003:**
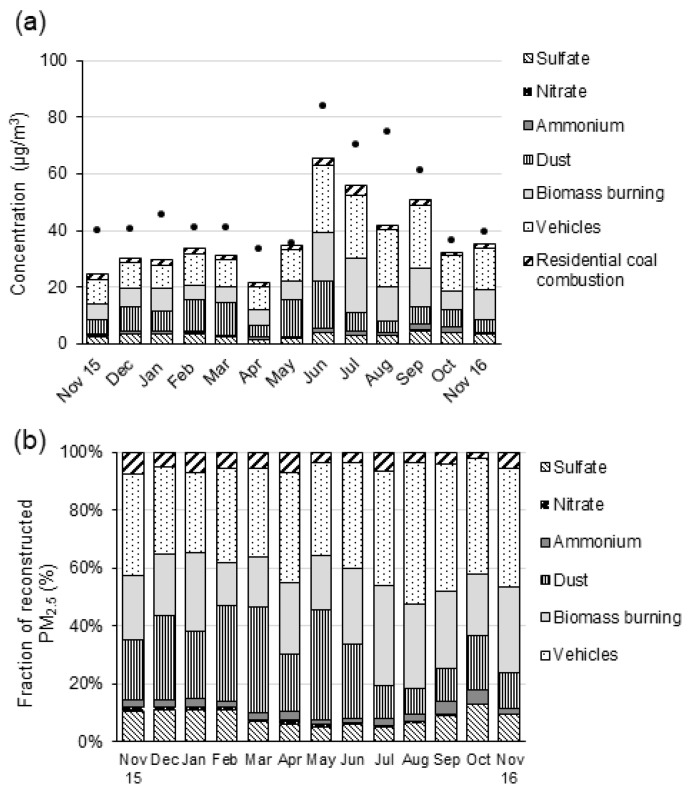
PM_2.5_ mass sources in units of (**a**) µg m^−3^ and (**b**) percent of reconstructed (measured/apportioned) PM mass. Fuel oil is not shown due to small contributions.

## Data Availability

Published data will be available shared for users via the Eastern Africa GEOHealth hub webpage subject to the data sharing policy.

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
