# Peer review of "Source Apportionment of Fine Organic Particulate Matter (PM2.5) in Central Addis Ababa, Ethiopia"

_ijerph, 2021, doi:10.3390/ijerph182111608_

Round 1
Reviewer 1 Report
As the title suggest the manuscripts details the results of a chemical speciation and source apportionment study of fine Particulate Matter (PM2.5) in Central Addis Ababa, Ethiopia. The study is well planned; however, a bit more information is needed in the manuscript before publication, and some improvements in the grammar.
Main Comments
- More information is needed as to how the source composition of the source profiles, for the chemical mass balance model, were selected and how suitable they were for local conditions. For example, on lines 289 – 290 it is stated: “In addition, sources including sulfate, nitrate, and ammonium concentrations input into CMB.” No information is given to what these sources looked like. I apologise if this information was available in the supplementary information.
Minor Comments:
- Line 44: “as only a few source apportionment studies undertaken to-date”, might read better as “as only a few source apportionment studies have been undertaken to-date”.
- Line 46: “shown” might read better as “has shown”.
- Lines 64-65: “. Only two source apportionment studies carried out in the city so far, but 64 none recently in today’s context”. This sentence needs to be re-worded.
- Line 93: “All quartz filters analyzed for elemental carbon (OC) and elemental carbon (EC)”, the first “elemental”, should be “organic”.
- Line 101-102: “The sampling approach and the settings described in more details on Tefera et al. 101 (2020). X-ray fluorescence used for elemental analyses.”, could read better as “The sampling approach and the settings were described in more details in Tefera et al. 101 (2020). X-ray fluorescence was used for elemental analyses.”
- Lines 105-106: “Schauer (2003) described the details on the methods of EC and OC analyses (Schauer, 2003). As described in detail by Tefera et al. (2020) (Tefera et al., 2020),”. Should be “Schauer (2003) described the details on the methods of EC and OC analyses. As described in detail by Tefera et al. (2020),”
- Line 169: “showed the most highs and lows of concentrations during”, might read better as “showed the highest and lowest of concentrations during”.
Author Response
Dear Reviewer 1,
Thank you for taking your valuable time to review this manuscript. We have addressed the comments and corrected typological suggestions.
Kindly find attached the responses in a tabular form from the authors.
Best,
Worku T.
Corresponding author

Reviewer 2 Report
Please note that the present article is not aligned with the journal's aims & scopes: https://www.mdpi.com/journal/ijerph/about
Please see more comments in the attachment:

Author Response
Dear Reviewer 2,
Thank you for taking your invaluable time to review this manuscript. The authors corrected the suggested typos and main comments.
Kindly find the attached authors' response file.
Best,
Worku T.
Corresponding author

Reviewer 3 Report
This study investigated the seasonal variation in fine particulate matter (PM2.5) and its chemical compositions in Addis Ababa, Ethiopia. The PM2.5 source apportionment was further conducted using a chemical mass balance (CMB) receptor model. This study revealed that PM2.5 in Addis Ababa was mainly from vehicular sources, biomass burning, and soil dust, and these three types of sources varied seasonally with higher contribution from vehicular sources and biomass burning in wet season and from soil dust in small rainy season. However, some concerns listed below need to be carefully considered. Meanwhile, more clarifications and discussions are necessary before the manuscript could be published in final format.
- In this study, only five or six PM5 samples in each month were collected and analyzed. How to ensure the representativeness of these samples? Could these samples be influenced by some specific polluted events? Or some polluted events were missed during the non-sampling period?
- This study suggested that vehicle emission is a major source of PM5. However, I am confused why concentrations of nitrates mainly from vehicle emission were very low. This result is contrary to the high contribution from vehicle emission.
- L93: OC should be organic carbon.
- References are repeated in some sentences, such as L105-106.
- I did not find the supplementary file in the website of the supplementary material.
- Please unify the format of units, such as a space is needed after numbers.
- L198-199: Please add the variation in ambient temperature.
- L214: In the field of atmospheric chemistry, soot is usually referred to black carbon. So, the “soot” in this sentence is easily misunderstood.
- Why most of measured OC concentrations were lower than the sum of OC from four sources during June-November 2016 period in Figure 2a?
- L265-269: In colder seasons, residential heating emission does contribute large amounts of primary OC to TOC (Zhang et al., 2020), which can be referenced to support your results.
- L276: “shows” should be “show”.
Zhang, J., Liu, L., Xu, L., Lin, Q., Zhao, H., Wang, Z., Guo, S., Hu, M., Liu, D., Shi, Z., Huang, D., and Li, W.: Exploring wintertime regional haze in northeast China: role of coal and biomass burning, Atmos. Chem. Phys., 20, 5355-5372, https://doi.org/10.5194/acp-20-5355-2020, 2020.
Author Response
Dear Reviewer 3,
Thank you for taking your invaluable time to review this manuscript. The authors corrected the suggested typos and main comments.
Kindly find the attached authors' response file.
Best,
Worku T.
Corresponding author

Round 2
Reviewer 1 Report
I now only have only a few small suggestions:
Line 52: “problem, not well tracked” might be better as “problem is not well tracked”
Line 56: “studies has been” could read better as “studies have been”.
Line 77: I am not sure what was intended by <25>, <10>, <15> and <5>.
Line 81: “studies carried” could read better as “studies have been carried”
Line 168: What was [AL1]?
Line 411-412: “The ambient aerosols collected from November 2015 to November 2016 411 at a city center location in Addis Ababa city.” might read better as “Ambient aerosols samples were collected from November 2015 to November 2016 411 at a city center location in Addis Ababa city.”
Lines 413-414: “used in which molecular markers: EC|OC, hopanes, picene, 413 and levoglucosan, included in the model.” Might read better as “was used in which molecular markers: EC|OC, hopanes, picene, 413 and levoglucosan, were included in the model.”
Line 418: “aerosols, highly affected” might read better as “aerosols were highly affected”
Line 149: “aerosols, formed” may read better as “aerosols which are formed”
Author Response
Dear Reviewer 1:
Thank you for the invaluable comments and edits to improve the manuscript. We have addressed your comments and corrected the edits per suggestion as shown in the attached word document file.
Best,
Worku

Reviewer 2 Report
Please find further comments/queries in the .pdf file attached.

Author Response
Dear Reviewer 2:
Thank you again for the invaluable comments and edits to improve the manuscript. We have addressed your comments and corrected the edits as attached herewith.
Best,
Worku
